# The Clinical and Sonographic Features of Cervical Muscle Involvement in Patients with Frozen Shoulder: A Retrospective Study

**DOI:** 10.3390/biomedicines12102395

**Published:** 2024-10-19

**Authors:** Alice Chu-Wen Tang, Shih-Ting Huang, Szu-Yuan Wu, Simon Fuk-Tan Tang

**Affiliations:** 1Department of Physical Medicine and Rehabilitation, Fu Jen Catholic University Hospital, New Taipei City 24352, Taiwan; alicetang7716@gmail.com; 2Department of Physical Medicine and Rehabilitation, Lo-Hsu Medical Foundation, Lotung Poh-Ai Hospital, Yilan City 265501, Taiwan; p801003@gmail.com; 3Department of Radiology Oncology and Big Data Center, Lo-Hsu Medical Foundation, Lotung Poh-Ai Hospital, Yilan City 265501, Taiwan; szuyuanwu5399@gmail.com; 4Department of Physical Medicine and Rehabilitation, Linkou Chang Gung Memorial Hospital, Taoyuan 33305, Taiwan

**Keywords:** frozen shoulder, ultrasonography, compensatory mechanism, scalene muscle, levator scapulae muscle, shoulder ROM

## Abstract

**Background/Objectives**: Frozen shoulder is a common shoulder disorder that often places limitations on the range of motion of the shoulder. The disease may induce neck pain due to overuse of the neck muscle in an attempt to compensate for lack of shoulder movement. In clinical practice, swelling and inflammation of the scalene and levator scapulae may be detected via sonography in patients with frozen shoulder. The aim of this study was, therefore, to determine whether the involvement of the scalene complex or levator scapulae could compensate for the limited motion of the shoulder in patients with frozen shoulder. **Methods**: We retrospectively reviewed the medical records of 362 patients with unilateral frozen shoulder. These patients were divided into four groups depending on the involvement of the scalene complex or levator scapulae muscle. The range of motion of the shoulder—encompassing flexion, abduction, and external rotation—was measured with a goniometer. We also performed an ultrasound scan on each shoulder. The involvement of the scalene complex and levator scapulae muscle was also assessed via musculoskeletal ultrasound. **Results**: The range of motion of the shoulder in terms of flexion, abduction, external rotation, and total range of motion differed significantly between the four groups (*p* < 0.05). Patients in whom the scalene complex or levator scapulae muscle was involved demonstrated a significantly wider range of motion in different shoulder directions than patients without the involvement of those muscles (*p* < 0.05). **Conclusions**: A greater range of motion in the shoulder can be obtained through the activation of the scalene complex or levator scapulae muscle, which act to compensate for the lack of shoulder movement in patients with frozen shoulder. These two muscles showed thickening and hypoechoic changes upon sonography.

## 1. Introduction

Adhesive capsulitis, also termed frozen shoulder (FS), occurs in approximately 26% of the adult population and is clinically a common shoulder pain disorder [1]. This disease often occurs in middle-aged people between 40 and 60 years old, and women are four times more likely to suffer from this disease than men. It may present as sudden severe shoulder pain or an insidious pain symptom alongside limited motion in the shoulder joint [2]. It generally first affects the external rotation of the shoulder, later affecting abduction; it can also cause limitations in the functioning of the patient’s shoulder, especially during overhead movements (e.g., hanging clothes) and when the shoulder crosses the midline to the other side (e.g., when putting on a seatbelt). Moreover, frozen shoulder adversely effects patients’ quality of life; affected individuals may not be able to take a shower, comb their hair, or put on clothes independently.

Although a comprehensive understanding of frozen shoulder is still lacking, efforts to fully reveal the pathophysiology of frozen shoulder are ongoing. The disease has been found to cause a persistent inflammatory response that leads to fibrosis of the muscles, ligaments, and joint capsules in the shoulder. Research involving magnetic resonance imaging of frozen shoulder has shown that the anatomical change mainly occurs through the thickening of the coracohumeral ligament [3,4,5], joint capsule, rotator cuff interval [5,6], and axillary fold [6,7]. These major structural alterations can also be detected through ultrasound [8,9]. On the other hand, studies in molecular biology have indicated the increased presence of inflammatory mediators such as the interleukins IL-1β, IL-6, IL-8, and tumor necrosis factor-alpha (TNF-α) and matrix metalloproteinases (MMPs) in the joint capsule and subacromial bursa [10,11,12]. 

The intimate anatomical relationship and innervation between the neck and shoulder have long been underestimated in studies of frozen shoulder. Nevertheless, the integration of the neck and shoulder has attracted significant attention in the field of head and neck surgery, and shoulder dysfunction following radical neck dissection was first documented in 1952. Later, a cluster of symptoms—including shoulder pain, limited shoulder abduction, and scapula winging resulting from damage to the spinal accessory nerve—were further described as “shoulder syndrome” [13]. Studies in shoulder kinematics tend to acknowledge the link between muscle activation and motor control in the shoulder and upper back. Constraint placed upon the range of motion of the shoulder is also related to the scapulohumeral rhythm [14]. Some studies have shown that subjects with frozen shoulder exhibit an imbalanced and hyperactive upper and lower trapezius when the shoulder is in various elevated positions [15,16,17]; however, other studies have observed less pronounced activity of the upper and lower trapezius [18]. These contradictory findings might suggest different compensatory strategies from patient to patient. Due to the anatomical relationship between the trapezius and levator scapulae and the similar direction of their muscle fibers, crosstalk via electromyography signals from the levator scapulae may have contributed to the electromyography findings related to the upper trapezius [18].

Moreover, decreased activity in the infraspinatus tendon during scaption and over-activity of the pectoralis major during a thumb-to-waist task have also been noted [18]. Despite image-based evidence of fibrotic change in non-contractile tissue such as capsules and ligaments [5], the importance of contractile tissue has been neglected [19]. Few studies have focused on changes in muscle control in frozen shoulder. One study found that the posterior deltoid, infraspinatus, and teres minor exhibited muscle stiffness when patients with frozen shoulder performed internal rotation [20]. These results suggest that an interaction between muscle condition and shoulder range of motion cannot be ruled out.

Among the frequently described symptoms of frozen shoulder, patients frequently complain of neck pain, but it receives relatively little clinical attention. Overuse of the neck muscle might serve to compensate for loss of shoulder movement [2]. In our clinical practice, we have found that swelling and inflammation of the scalene complex and levator scapulae can be detected by sonography in some patients with frozen shoulder. To our knowledge, these findings have not been documented in previous studies. Changes in motor control and muscle recruitment in patients with neck–shoulder pain performing functional tasks have been described in many studies [21,22,23,24]; therein, both upper trapezius and sternocleidomastoid muscle overactivity were observed, but the features of the scalene complex muscle were left unexplored. One study in elite swimmers suffering from unilateral shoulder pain found that the anterior scalene muscle undertook greater activity during functional tasks [25].

Overall, compensatory strategies are important in establishing a panoramic view of frozen shoulder, but limited studies have reported relevant findings (especially concerning the role of the levator scapulae and scalene complex). Therefore, the purpose of this study was to determine whether the involvement of the scalene complex or levator scapulae could compensate for the narrow range of shoulder motion in patients with frozen shoulder. Our hypothesis was that patients utilizing the scalene complex or levator scapulae as a compensatory strategy have a greater range of motion in their shoulder.

## 2. Materials and Methods

### 2.1. Subject

Patients with unilateral shoulder pain and restricted range of motion were recruited from outpatient clinics. Our study was approved by the institutional review board of a tertiary medical center, Tzu Chi Hospital, on 15 July 2022, with approval code: IRB 111-137-B. Our inclusion criteria were as follows: (1) unilateral shoulder pain lasting more than 1 month, (2) restriction of shoulder range of motion in the coronal, sagittal, and frontal plane, and (3) age > 20 years old. The exclusion criteria were as follows: (1) patients with bilateral frozen shoulders, (2) upper limb or shoulder fracture, (3) documented rotator cuff tear, and (4) individuals with shoulder surgery, infection, rheumatic disease, and malignancy.

This retrospective study included 362 patients who were diagnosed with frozen shoulder between September 2019 and September 2021 at our hospital. The cohort included 159 males and 203 females with an average age of 58.2 ± 13.2 (from 25 to 99 years old). These patients were divided into four groups according to the involvement of the scalene complex or levator scapulae. Frozen shoulder without involvement of the scalene complex and levator scapulae was denoted Group 1 (N = 254), versus frozen shoulder with involvement of the scalene complex (Group 2, N = 53), or levator scapulae (Group 3, N = 46), and frozen shoulder with involvement of both muscles (Group 4, N = 9). Demographic data, including sex, age, smoking, alcohol consumption, associated risk factors (hyperlipidemia, hypertension, hyperglycemia, thyroid disease, gout, and rheumatic arthritis), and shoulder range of motion (ROM) were obtained from the patient’s electronic medical records. Patients’ maximum painless range of shoulder motion (including flexion, abduction, and external rotation) was measured using a goniometer while the patient was seated with their clavicle and scapular fixed.

### 2.2. Ultrasound Examination

Shoulder ultrasound examinations were conducted according to standard protocol 9 and using an ultrasound machine (Philips Affiniti 50G, Philips, Eindhoven, The Netherlands) equipped with a probe frequency range between 5 and 12 MHz. The same senior ultrasound operator performed all ultrasound examinations in this study, and the dynamic focus, gain, and time–gain compensation were all fixed. The ultrasound examination was performed by a single doctor who has been performing shoulder ultrasounds for more than 20 years. All the patients were examined by the same doctor in the follow-up.

To perform ultrasound examination, we stipulated that patients remain in the supine position in order to scan the bicep tendon with elbow flexion at 90 degrees. Afterward, the upper arm was externally rotated to check the subscapularis tendon. Then, we kept the patient in a sidewards lying position with their arm completely rotated internally, their elbow flexed, and their hand placed on the lower back in order to check the supraspinatus (SS) tendon and subdeltoid subacromial bursa. A round pillow was placed under the upper back for posture fixation. To evaluate the scalene complex, the patient was placed in the supine position, and the linear probe was placed at the sixth level of the cervical spine in order to observe the features of the anterior and posterior tubercle structures within the transverse process. The posterior tubercle was shorter than the anterior tubercle at C6. The scalene complex can be seen immediately below the subcutaneous layer. When the scalene complex was involved, hypoechoic changes and swelling were noted in the muscles (Figure 1). To observe the levator scapulae muscle, the linear probe of sonography was placed at the superior angle of the medial border of the scapula, pointing to the upper cervical spine. The involvement of this muscle was confirmed when the ultrasound showed hypoechoic change and swelling of the muscles (Figure 2). After confirming the importance of these two muscles, about 8 years ago, scalene muscles and levator scapulae were examined during routine shoulder US.

### 2.3. Statistical Analysis

Statistical analysis was performed with the use of SAS Software Version 9.4. The categorical variables are presented as counts and percentages, and continuous variables are presented. Comparisons of these clinical data between each group were made using the chi-square test for categorical variables and a one-way analysis of variance (ANOVA) for continuous variables. Using a linear regression model, we calculated the R square value of the regression model to assess the relationship between the total shoulder range of motion and other clinical data. Finally, a *p*-value of <0.05 was considered statistically significant.

## 3. Results

Patients’ characteristics, risk of adhesive capsulitis, and range of motion of shoulder are summarized in Table 1. Females were predominant within our cohort. The ROM values of the shoulder joint in terms of flexion, abduction, external rotation, and total ROM were 121 ± 28.82, 52 ± 13.88, 31 ± 17.34, and 203 ± 50.02 degrees, respectively, in Group 1; 144 ± 15.94, 61 ± 14.88, 40 ± 17.17, and 244 ± 36.29 degrees, respectively, in Group 2; 138 ± 19.75, 58 ± 12.17, 39 ± 13.44, and 236 ± 34.43 degrees, respectively, in Group 3; and 142 ± 28.17, 59 ± 15.57, 42 ± 19.53, and 242 ± 53.04 degrees, respectively, in Group 4. Significant differences in the range of motion of the shoulder in terms of flexion, abduction, external rotation, and total range of motion were noted between the four groups (*p* < 0.05) (Table 1). There was a significant difference between the total range of motion between Group 1 and Group 2 and between Group 1 and Group 3 (*p* < 0.05) (Figure 3). These results may indicate the involvement of either the levator scapulae or scalene complex to compensate for the lack of shoulder mobility in patients with frozen shoulder. The shoulder ROM in the group that featured both scalene complex and levator scapulae involvement was not significantly different from that of the other three groups. Furthermore, the actual mean angles of various shoulder movements in Group 4 were obviously larger than those in Group 1. The linear regression model disclosed that the involvement of the scalene complex or/and levator scapulae, age over 65 years old, and MACC had significantly positive associations with total range of motion of the shoulder joint (Table 2).

## 4. Discussion

The present study aimed to determine whether the involvement of the scalene complex or levator scapulae could compensate for the restricted range of shoulder motion characteristic of frozen shoulder. In this retrospective study, adhesive capsulitis with the involvement of the scalene complex or levator scapulae muscle allowed for greater shoulder ROM than adhesive capsulitis without scalene complex and levator scapulae involvement. The range of motion of the shoulder in flexion, abduction, and external rotation in subjects with the involvement of the scalene complex or levator scapulae was greater than in subjects without scalene and levator scapulae involvement. Such an observation may be due to the alternation of scapular kinematics, which serves to compensate for shoulder elevation. The ROM of subjects with both scalene complex and levator scapulae muscle involvement did not improve; we might attribute this observation to our limited number of participants and corresponding low statistical power. The actual mean angles of different shoulder movements in this group were obviously larger than those of Group 1 (frozen shoulder without scalene complex and levator scapulae involvement).

When shoulder range of motion is limited due to pain or a form of shoulder disorder, the neck muscle and scapular muscle will become involved and improve shoulder motion via compensatory mechanisms [2,26]. The findings from Rundquist PJ et al. [26] support the theory of scapular compensation after the loss of glenohumeral ROM for the purpose of achieving greater humerus-to-trunk scapular plane elevation. Vermeulen et al. [27] also demonstrated the increased contribution of the scapular muscles to scapulohumeral rhythm in subjects with frozen shoulder.

Previous studies have indicated that the levator scapulae muscle is activated upon shoulder elevation [28]. De Freitas et al. [29] showed that in their EMG study, the levator scapulae had great involvement in abduction and elevation, moderate involvement in shoulder flexion, and minimal involvement in scapular retraction and shoulder extension. Behrsin JF [4] also found that levator scapulae activity was increased in the outer range of flexion and abduction in some subjects. Castelein et al. [30] showed higher levator scapulae activity in elevation with external rotation without additional load. With additional load, both scaption and elevation with external rotation involved more significant muscle activity. Although there was no significant difference found in the activity of the levator scapulae between patients with subacromial impingement syndrome and healthy controls in the study by Castelein et al. [31], their results related to subacromial impingement syndrome rather than adhesive capsulitis. The purpose of the levator scapulae involvement was primarily to elevate the scapula; thus, we speculated from our study that the levator scapulae may be activated to elevate the shoulder and compensate for the loss of shoulder motion to a certain degree, in patients with adhesive capsulitis. Ferlito R. et al. [32] indicated that three type of interventions will be effective for frozen shoulder as follows: (1) manual therapy in patients with subacromial impingement, (2) therapeutic exercise programs including interventions on the scapulothoracic complex in patients with subacromial impingement syndrome, and (3) therapeutic exercise programs including interventions on the scapulothoracic complex in patients with frozen shoulder. Recently, with the advancement of interventional musculoskeletal ultrasound, those lesions in frozen shoulder can be detected by ultrasound. Therefore, echo-guided injections to those lesions became possible with steroids or other agents such as platelet rich plasm. Exercise programs can be started after inflammation has been well controlled.

On the other hand, the scalene muscles originate from upper cervical spine and are attached at the first and second ribs. This group of muscles are also known as the respiratory accessory muscles, which can elevate the anterior chest wall. Activation of these muscles has been significantly associated with shoulder pain and limited joint movement. In the current study, hypoechoic change and swelling of the scalene muscle as detected via soft tissue sonography were found in patients with shoulder pain and a limited range of shoulder motion. Similarly, Hidalgo-Lozano [25] mentioned that subjects with unilateral shoulder pain demonstrate higher scalene muscle activity with increased surface EMG signals during and after performing a functional upper-limb task. Although these findings were found in swimmers with unilateral shoulder pain, the results indicate hyperactivity of the scalene muscle during functional motor tasks in subjects with unilateral shoulder pain. Madeleine [24] has also observed that in patients with neck pain, hyperactivity signals of the scalene muscle appeared upon surface electromyography both during and after the performance of a functional task. Furthermore, the swelling of the scalene muscles might cause nerve entrapment of the C5 and C6 spinal nerves, which can penetrate between the anterior scalene and middle scalene to the brachial plexus. Some patients with frozen shoulder experienced numbness in their upper limb, which was reminiscent of radiculopathy. We should conduct further studies to characterize and verify the relationship between muscle swelling and radiculopathies.

A clear picture of the condition of frozen shoulder cannot be gleaned without a discussion of microenvironmental changes. The disease is characterized by the progress of inflammation, proliferation, angiogenesis, and fibrosis [33,34]. Oxidative stress caused by hypoxia or infection and chronic low-grade inflammation followed by the subsequent angiogenesis and fibrosis can all perpetuate frozen shoulder [35]. Apart from the famous inflammatory cascade mediated by the transcription nuclear factor Kappa B (NF- kB) [36] and the subsequently released cytokines (such as IL-1, IL-6, and TNF-α [37,38,39]), the adhesion molecule ICAM-1 [37] and alarmins, which are endogenous molecules released after non-programmed cell death, are also crucial in the development of frozen shoulder. The alarmins high-mobility group box 1 (HMGB1), IL-33, S100A8, and S100A9 have been found to be elevated in frozen shoulder joint capsules [40]. HMGB1 was shown to regulate inflammation and immune response [41], while IL-33 was found to be related to angiogenesis and fibrosis [35]. S100 altered the ratio of matrix metalloproteinase (MMPs) to tissue inhibitor metalloproteases (TIMPs), resulting in imbalanced turnover of the extracellular matrix. Furthermore, a recent study has also found that the increased presence of S100 in frozen shoulder might change perceptions of pain [42]. Notably, current bench studies emphasize molecular expression in only the shoulder joint capsule. The fact that both non-contractile and contractile tissues encounter oxidative stress is somewhat inconspicuous. In this study, we concluded that the levator scapulae and scalene complex compensated for the lack of mobility in frozen shoulder, and their excessive use might cause inflammatory changes in both muscles. However, the muscles surrounding the shoulder joint might also be affected by adjacent cytokine release and changes in the microenvironment. Further studies are needed to confirm this.

## 5. Limitation

The several limitations of this study should be acknowledged. First and foremost, the sample size was relatively small, which inherently limits the statistical power. Second, the study was conducted at a single center under the guidance of a single physiatrist, further constraining its generalizability to broader clinical settings. Further research with a larger patient population or multi-center study are warranted to enhance the reliability and applicability of our findings.

## 6. Conclusions

The levator scapulae and scalene complex can clearly compensate for the constraints placed on the shoulder range of motion in patients with frozen shoulder. This compensatory mechanism may further improve patients’ shoulder function and quality of life.

## Figures and Tables

**Figure 1 biomedicines-12-02395-f001:**
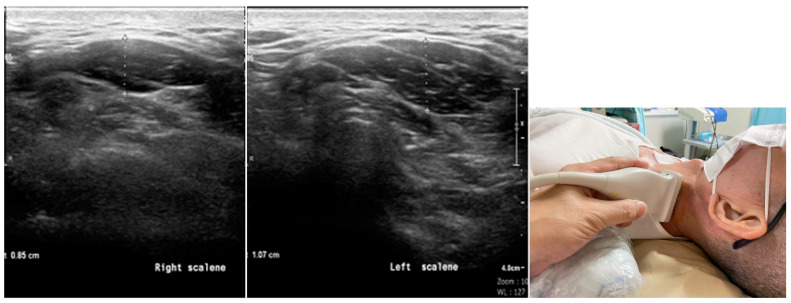
Sonogram shows hypoechoic change in the scalene complex with increased thickness on the left side. Normal appearance of the scalene complex on the right side. Position of the probe is shown in the smaller frame on the right-hand side.

**Figure 2 biomedicines-12-02395-f002:**
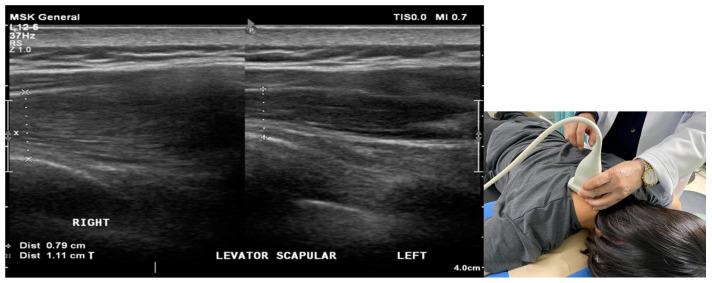
Sonogram shows the levator scapulae muscle with increased thickness (1.65 cm) and hypoechoic change at the supraspinal fossa on the right side. Normal appearance and thickness (1.28 cm) of the levator scapulae muscle on the left side. Position of the probe is shown in the smaller frame on the right-hand side. +·····+ represent thickness of muscle.

**Figure 3 biomedicines-12-02395-f003:**
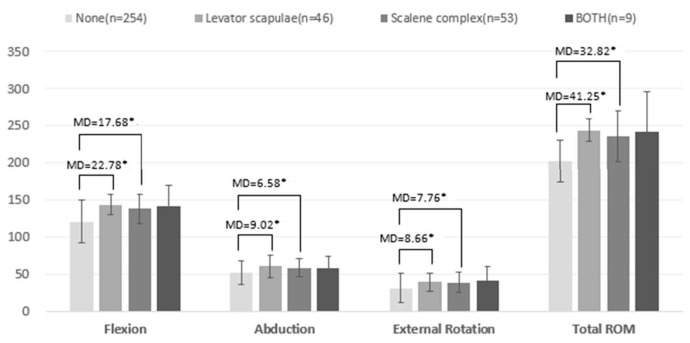
Shoulder range of motion in flexion, abduction, and external rotation in subjects featuring involvement of the scalene complex or levator scapulae was greater than that in subjects without scalene and levator scapulae involvement. * = *p* < 0.05.

**Table 1 biomedicines-12-02395-t001:** Patients’ characteristics, risk of adhesive capsulitis, and shoulder range of motion.

	Non (=254)		Levator Scapulae		Scalene Complex		Both (=9)		*p*-Value
			(=46)		(=53)				
	N	%	N	%	N	%	N	%	
**Sex**									0.8875
FEMALE	140	55.10%	27	58.70%	30	56.60%	6	66.70%	
MALE	114	44.90%	19	41.30%	23	43.40%	3	33.30%	
**Age group**									0.791
AGE < 50	61	24.00%	13	28.30%	19	35.80%	3	33.30%	
AGE 51–65	104	40.90%	18	39.10%	20	37.70%	4	44.40%	
AGE 66–85	75	29.50%	13	28.30%	11	20.80%	1	11.10%	
AGE > 85	14	5.50%	2	4.30%	3	5.70%	1	11.10%	
**Smoking**	24	9.40%	2	4.30%	4	7.50%	0	0.00%	0.5238
**Alcohol**	9	3.50%	2	4.30%	2	3.80%	0	0.00%	0.9365
**Rheumatoid arthritis (RA)**	4	1.60%	0	0.00%	2	3.80%	0	0.00%	0.4349
**Thyroid problems**	3	1.20%	3	6.50%	4	7.50%	0	0.00%	0.0227
**DM**	41	16.10%	1	2.20%	7	13.20%	2	22.20%	0.0784
**Gout**	16	6.30%	2	4.30%	1	1.90%	1	11.10%	0.5133
**Hypertension**	71	28.00%	8	17.40%	11	20.80%	2	22.20%	0.3793
**Hyperlipidemia**	57	22.40%	7	15.20%	10	18.90%	2	22.20%	0.7059
**MACC**	15	5.90%	1	2.20%	4	7.50%	1	11.10%	0.5993
**Spondylosis**	51	20.10%	11	23.90%	13	24.50%	1	11.10%	0.7306
**COPD**	32	12.60%	9	19.60%	2	3.80%	1	11.10%	0.1161
	**MEAN**	**SD**	**MEAN**	**SD**	**MEAN**	**SD**	**MEAN**	**SD**	
**Flexion**	121	28.82	144	15.94	138	19.75	142	28.17	<0.0001
**Abduction**	52	13.88	61	14.88	58	12.17	59	15.57	<0.0001
**External rotation**	31	17.34	40	17.17	39	13.44	42	19.53	0.0003
**Total ROM**	203	50.02	244	36.29	236	34.43	242	53.04	<0.0001

Abbreviations: DM: Diabetes Mellitus, MACC: Major Adverse Cardiovascular Events, COPD: Chronic Obstructive Pulmonary Disease.

**Table 2 biomedicines-12-02395-t002:** Relationship between total shoulder ROM and other clinical data.

Total_ROM		R^2^	19.3%		
	REF	Estimated Value	95%CI		*p*-Value
Group 2: Scalene complex	**Group 1**	32.3906	18.7889	45.9923	**<** **0.0001**
Group 3: Levator scapulae		38.2534	23.876	52.6307	**<** **0.0001**
Group 4: Both		38.9318	8.9646	68.8989	**0.0109**
MALE	FEMALE	−4.3558	−14.5771	5.8655	0.4036
AGE > 85	AGE < 50	−35.8777	−59.8045	−11.951	**0.0033**
AGE 65–85		−17.5805	−31.2241	−3.9368	**0.0116**
AGE 50–64		−8.8953	−20.9484	3.1577	0.148
Smoking	NO	−7.042	−26.4535	12.3695	0.4771
Alcohol	NO	−9.6668	−36.7555	17.422	0.4843
Thyroid problems	NO	4.4709	−24.7098	33.6515	0.764
DM	NO	−9.493	−26.0829	7.0969	0.2621
Gout	NO	0.9952	−21.2726	23.263	0.9302
Hypertension	NO	0.946	−12.0346	13.9266	0.8864
Hyperlipidemia	NO	14.9299	−0.4874	30.3473	0.0577
Rheumatoid arthritis	NO	−13.2946	−43.764	17.1748	0.3925
MACC	NO	−24.9725	−46.5906	−3.3544	**0.0236**
Spondylosis	NO	0.8853	−11.436	13.2066	0.888
COPD	NO	6.5472	−8.1451	21.2395	0.3824

Abbreviations: DM: Diabetes Mellitus; MACC: Major Adverse Cardiovascular Events; COPD: Chronic Obstructive Pulmonary Disease.

## Data Availability

The original contributions presented in the study are included in the article, further inquiries can be directed to the corresponding author.

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
