# Peer review of "The Clinical and Sonographic Features of Cervical Muscle Involvement in Patients with Frozen Shoulder: A Retrospective Study"

_biomedicines, 2024, doi:10.3390/biomedicines12102395_

Round 1

Reviewer 1 Report

Comments and Suggestions for Authors

Dear authors, 

A very original and beautiful work. We have some areas where we need to improve.  

I can't wait to see it again after the edits.

The Clinical and Sonographic Features of Cervical Muscles Involvement in Patients with Frozen Shoulder: A Retrospective Study

Abstract:

Background/Objectives:

..... shoulder movement. ;      ......for shoulder movement.

.... swollen and;  ..... swelling and

Methods:

The medical record/s.............  shoulder/s ................... reviewed.

.........external rotation were;   .................external rotation was...

Results:

........of motion were significantly......;  ............of motion was significantly...

......... those muscle involvement (p<0.05).;..... muscle involvement (p<0.05).;

Conclusions:

Greater range of motion of shoulder can....; A greater range of motion of shoulder/s  can......

1. Introduction

English editing by a native English speaker is required.

..........................as frozen shoulder (FS);   ........frozen shoulder (FS)...

.................... in revealing; ................... to reveal...

Researches of...................; Research of .....

................winging resulted ; ....winging,  resulting

Few researches ;  Few types of research

These results suggested interaction; These results suggested that the interaction

...............compensate strategy for the loss of shoulder movement [2].; .....compensation strategies for ....

While/Meanwhile, features of scalene complex muscle were leaved/left unexplored.

2. Materials and Methods

2.1. Subject

- Please let's indicate the number and date of ethics committee approval.

- Inclusion criteria were as followings: The inclusion criteria were as follows:

- "Was there a specific age group included in the criteria for participation?" (Children/elders etc.).

- Exclusion criteria were as followings:   The exclusion criteria were as follows:

- Were individuals with shoulder surgery, infection, rheumatic disease, malignancy, or chronic diseases such as diabetes and thyroid disease included or excluded from the study?

- This retrospective study included 362 ??? patients. (The total of the groups is 363).

- ..............from the patients' electronic... ; .......from the patient's electronic..... 

- ............while patient seated with.......; ........while the patient was seated.....

2.2. Ultrasound Examination

- Could you please provide information about the number of years the doctor has been performing shoulder ultrasounds?

- ....the SS ???? tendon....  (Let's write it short using parentheses).

- Let's add small boxes next to Figures 1 and 2 to show the areas where we placed the probe.

2.3. Statistical Analysis

- In some outcomes, such as age (mean ± SD) was not used? (SD; Standard deviation)

3. Results

- Female was predominant.;  Females were predominant.

- Table 1. Patients’ characteristics, risk factor of ..., ; Patients’ characteristics, the risk factors of ...

- Table 1.    The regulation of age/years  should be considered;

Age<50,

65<Age>51,

85<Age>66,           Please make sure to arrange the table accordingly.

Age>80.

- Let's write the abbreviations below the table. (DM, MACC, COPD......).     Let's write SD instead of std.

- 144±15.84, let's correct it. (144±15.94)

- Table 2. Fix age.

- Table 2. What does Amoking mean?

- Table 2. Let's write the abbreviations below the table. (DM, MACC, COPD......)

4. Discussion

- The findings from Rundquist P. J. group [10].....; The findings from Rundquist et al. [10]

- Vermeulen [11] et al; Vermeulen et al. [11]

- Previous studies have indicated that levator scapulae is activated upon shoulder elevation.  (Let's add a references).

- The discussion focuses more on pathophysiology than on the US. Let's reorganize it by emphasizing more emphasis on US studies.

5. Limitation

6. Conclusion

- Are your clinic's scalene muscles and levator scapulae examined during routine shoulder-US? (Because the study is retrospective).

- The similarity rate is 37%.  It should be reduced.

7. Referencess

References must be up to date.

References need to be rearranged.

Comments on the Quality of English Language

English editing is needed.

Author Response

Comment 1: Abstract- Background/Objectives:

..... shoulder movement. ;      ......for shoulder movement.

.... swollen and;  ..... swelling and

Response:

Thank you! We have made amendment.

Comment 2: Abstract- Methods:

The medical record/s.............  shoulder/s ................... reviewed.

.........external rotation were;   .................external rotation was...

Response:

Thank you! We have made amendment.

Comment 3: Abstract- Results:

........of motion were significantly......;  ............of motion was significantly...

......... those muscle involvement (p<0.05).;..... muscle involvement (p<0.05).;

Response:

Thank you! We have made amendment.

Comment 4: Abstract- Conclusions:

. Greater range of motion of shoulder can....; A greater range of motion of shoulder/s  can......

Response:

Thank you! We have made amendment.

Comment 5: Introduction

English editing by a native English speaker is required.

..........................as frozen shoulder (FS);   ........frozen shoulder (FS)...

.................... in revealing; ................... to reveal...

Researches of...................; Research of .....

................winging resulted ; ....winging,  resulting

Few researches ;  Few types of research

These results suggested interaction; These results suggested that the interaction

...............compensate strategy for the loss of shoulder movement [2].; .....compensation strategies for ....

While/Meanwhile, features of scalene complex muscle were leaved/left unexplored.

Response:

Thank you! English editing had completed.

Comment 6: Materials and Methods- Subject

- Please let's indicate the number and date of ethics committee approval.

Response:

Thank you! This study was approved by the Research and Ethics Committee of Tzu Chi Hospital on July 15, 2022 with approval code: IRB 111-137-B. We have reported the number and date of ethics committee approval for this study in the revised manuscript.

Comment 7: Materials and Methods- Subject

- Inclusion criteria were as followings: The inclusion criteria were as follows:

Response:

Thank you. We made amendment.

Comment 8: Materials and Methods- Subject

- "Was there a specific age group included in the criteria for participation?" (Children/elders etc.).

Response:

Thank you! We add this age criteria in the materials and methods. We recruited patients above 20 years old.

Comment 9: Materials and Methods- Subject

- Exclusion criteria were as followings:   The exclusion criteria were as follows:

Response:

Thank you. We made amendment.

Comment 10: Materials and Methods- Subject

Were individuals with shoulder surgery, infection, rheumatic disease, malignancy, or chronic diseases such as diabetes and thyroid disease included or excluded from the study?

Response:

Thank you!

Individuals with shoulder surgery, infection, rheumatic disease, malignancy were excluded from this study.

Comment 11: Materials and Methods- Subject

This retrospective study included 362 ??? patients. (The total of the groups is 363).

Response:

This retrospective study included 362 patients. We made a typo in the 2.1 Subject, (Group 2, N=53 instead of N=54). Therefore, the total of the groups is 362.

Comment 12: Materials and Methods- Subject

- ..............from the patients' electronic... ; .......from the patient's electronic..... 

- ............while patient seated with.......; ........while the patient was seated.....

Response:

Thank you. We made amendment.

Comment 13: Materials and Methods- Ultrasound Examination

Could you please provide information about the number of years the doctor has been performing shoulder ultrasounds?

Response:

The doctor has been performing shoulder ultrasounds for more than 20 years.

Comment 14: Materials and Methods- Ultrasound Examination

- ....the SS ???? tendon....  (Let's write it short using parentheses).

Response:

Thank you. We made amendment.

Comment 15: Materials and Methods- Ultrasound Examination

- Let's add small boxes next to Figures 1 and 2 to show the areas where we placed the probe.

Response:

Yes, we add the position of probe next to Figures 1 and 2.

Comment 16: Materials and Methods- Statistical Analysis

- In some outcomes, such as age (mean ± SD) was not used? (SD; Standard deviation)

Response:

Thank you. We made amendment.

Comment 17: Results

- Female was predominant.;  Females were predominant.

- Table 1. Patients’ characteristics, risk factor of ..., ; Patients’ characteristics, the risk factors of ...

Response:

Thank you. We made amendment.

Comment 18: Results

- Table 1.    The regulation of age/years  should be considered;

Age<50,

65<Age>51,

85<Age>66,           Please make sure to arrange the table accordingly.

Age>80.

Response:

Thank you. We made amendment.

Comment 19: Results

- Let's write the abbreviations below the table. (DM, MACC, COPD......).     Let's write SD instead of std.

Response:

Thank you. We made amendment.

Comment 20: Results

- 144±15.84, let's correct it. (144±15.94)

Response:

Thank you. We made amendment.

Comment 21: Results

- Table 2. Fix age.

Response:

Thank you. We made amendment.

Comment 22: Results

- Table 2. What does Amoking mean?

Response:

Thank you. We made amendment.

Comment 23: Results

- Table 2. Let's write the abbreviations below the table. (DM, MACC, COPD......)

Response:

Thank you. We made amendment.

Comment 24: Discussion

- The findings from Rundquist P. J. group [10].....; The findings from Rundquist et al. [10]

- Vermeulen [11] et al; Vermeulen et al. [11]

Response:

Thank you. We made amendment.

Comment 25: Discussion

- Previous studies have indicated that levator scapulae is activated upon shoulder elevation.  (Let's add a references).

Response:

We added the reference.

Comment 26: Discussion

- The discussion focuses more on pathophysiology than on the US. Let's reorganize it by emphasizing more emphasis on US studies.

Response:

We added some more description on US studies

Comment 27: Conclusion

- Are your clinic's scalene muscles and levator scapulae examined during routine shoulder-US? (Because the study is retrospective).

Response:

After we found the importance of these two muscles about 8 years ago. Scalene muscles and levator scapulae were examined during routine shoulder-US.

Comment 28: Conclusion

- The similarity rate is 37%.  It should be reduced.

Response:

Thank you. We reduced the similarity in the revised manuscript.

Comment 29: Reference

References must be up to date.

References need to be rearranged.

Response:

Thank you. We made amendment.

Comment 30:

Comments on the Quality of English Language

English editing is needed.

Response:

Thank you. We made amendment.

Reviewer 2 Report

Comments and Suggestions for Authors

The manuscript is a retrospective study aimed to determine if the involvement of scalene complex or levator scapulae can compensate for limited motion of the shoulder in patients with frozen shoulder. The medical record of 362 patients with unilateral frozen shoulder were retrospectively reviewed. All these patients were divided into four groups depending on the involvement of scalene complex or levator scapulae muscle. The shoulder range of motion including flexion, abduction and external rotation were measured with goniometer. In the meanwhile, ultrasound scanning of the shoulder was performed. The involvement of scalene complex and levator scapulae muscle was also checked by the musculoskeletal ultrasound

I read the article with interest, the title is well thought out and faithfully reflects the content of the study.

The abstract is adequately developed, and it is useful to frame the purpose of the study.                                                                                                                                                                                       

In the introduction, the characteristics of frozen shoulder have been shortly described. The materials and methods have been adequately described. The discussion is sufficiently developed, even if a little too synthetic.

Nevertheless, some minor changes are needed to be considered suitable for publication.

Comment 1: In the materials and method: please clearly indicate the inclusion and exclusion criteria in this study

Comment 2:  In the materials and method: what kind of rehabilitation was used? it always the same?

Comment 3: In the materials and method: Who did you perform in the follow-up of these patients? Still the same doctor?

Comment 4: In the discussion: It would be advisable to deepen the concept of rehabilitation, with information about the various contexts in which the different Protocols are used. Adding appropriate bibliographic references, such as: (Ferlito R. et al (2023) "Effectiveness of therapeutic interventions on the scapulothoracic complex in the management of patients with subachromatic splicing and frozen shoulder: a systematic review. ").

Comment 5: In the discussion: It would be appropriate to summarise the limitations of the study and include them in the same section of the discussion

Comment 6: Finally, additional English editing is needed. The Non-Native Speakers of English Editing Certificate was not signed.

Comments on the Quality of English Language

English editing is needed. The Non-Native Speakers of English Editing Certificate was not signed.

Author Response

Comment 1: In the materials and method: please clearly indicate the inclusion and exclusion criteria in this study

Response:

Thank you. We made amendment.

Comment 2:  In the materials and method: what kind of rehabilitation was used? it always the same?

Response:

In our setting, rehabilitation program included hot pack and interferential current for tissue softening and reduced pain followed with stretching, therapeutic and strengthening exercise.

Comment 3: In the materials and method: Who did you perform in the follow-up of these patients? Still the same doctor?

Response:

The follow-up of these patients was the same doctor.

Comment 4: In the discussion: It would be advisable to deepen the concept of rehabilitation, with information about the various contexts in which the different Protocols are used. Adding appropriate bibliographic references, such as: (Ferlito R. et al (2023) "Effectiveness of therapeutic interventions on the scapulothoracic complex in the management of patients with subachromatic splicing and frozen shoulder: a systematic review. ").

Response:

Thank you. We made amendment.

Comment 5: In the discussion: It would be appropriate to summarise the limitations of the study and include them in the same section of the discussion

Response:

Thank you. We made amendment.

Comment 6: Finally, additional English editing is needed. The Non-Native Speakers of English Editing Certificate was not signed.

Response:

Thank you. We made amendment.

Comment 7: Comments on the Quality of English Language

English editing is needed. The Non-Native Speakers of English Editing Certificate was not signed.

Response:

Thank you. We got the English Editing Certificate three days ago.

Round 2

Reviewer 1 Report

Comments and Suggestions for Authors

The similarity index is currently at 40%.

Let's work on reducing it.

Author Response

Comment 1: Abstract- Background/Objectives:

..... shoulder movement. ;      ......for shoulder movement.

.... swollen and;  ..... swelling and

Response:

Thank you! We have made amendment.

The disease may induce neck pain due to overuse of the neck muscle in an attempt to compensate for lack of shoulder movement. In clinical practice, swelling and inflammation of the scalene and levator scapulae may be detected via sonography in patients with frozen shoulder.

Comment 2: Abstract- Methods:

The medical record/s.............  shoulder/s ................... reviewed.

.........external rotation were;   .................external rotation was...

Response:

Thank you! We have made amendment.

We retrospectively reviewed the medical records of 362 patients with unilateral frozen shoulder.

The range of motion of the shoulder—encompassing flexion, abduction, and external rotation—was measured with a goniometer.

Comment 3: Abstract- Results:

........of motion were significantly......;  ............of motion was significantly...

......... those muscle involvement (p<0.05).;..... muscle involvement (p<0.05).;

Response:

Thank you! We have made amendment.

Patients in which the scalene complex or levator scapulae muscle was involved demonstrated a significantly wider range of motion in different shoulder directions than patients without the involvement of those muscles (p<0.05).

Comment 4: Abstract- Conclusions:

. Greater range of motion of shoulder can....; A greater range of motion of shoulder/s  can......

Response:

Thank you! We have made amendment.

A greater range of motion in the shoulder can be obtained through activation of the scalene complex or levator scapulae muscle, which act to compensate for the lack of shoulder movement in patients with frozen shoulder.

Comment 5: Introduction

English editing by a native English speaker is required.

..........................as frozen shoulder (FS);   ........frozen shoulder (FS)...

.................... in revealing; ................... to reveal...

Researches of...................; Research of .....

................winging resulted ; ....winging,  resulting

Few researches ;  Few types of research

These results suggested interaction; These results suggested that the interaction

...............compensate strategy for the loss of shoulder movement [2].; .....compensation strategies for ....

While/Meanwhile, features of scalene complex muscle were leaved/left unexplored.

Response:

Thank you! English editing had completed.

Adhesive capsulitis, also termed frozen shoulder (FS), occurs in approximately 26 % of the adult population and is clinically a common shoulder pain disorder [1].

Later, a cluster of symptoms—including shoulder pain, limited shoulder abduction, and scapula winging resulting from damage to the spinal accessory nerve—were further described as “shoulder syndrome” [13].

Few studies have focused on changes in muscle control in frozen shoulder.

Overuse of the neck muscle might serve to compensate for loss of shoulder movement [2]

therein, both upper trapezius and sternocleidomastoid muscle overactivity were observed, but the features of the scalene complex muscle were left unexplored.

Overall, compensatory strategies are important in establishing a panoramic view of frozen shoulder, but limited studies have reported relevant findings

Comment 6: Materials and Methods- Subject

- Please let's indicate the number and date of ethics committee approval.

Response:

Thank you! This study was approved by the Research and Ethics Committee of Tzu Chi Hospital on July 15, 2022 with approval code: IRB 111-137-B. We have reported the number and date of ethics committee approval for this study in the revised manuscript.

Our study was approved by the institutional review board of a tertiary medical center Tzu Chi Hospital on July 15,2022 with approval code: IRB 111-137-B.

Comment 7: Materials and Methods- Subject

- Inclusion criteria were as followings: The inclusion criteria were as follows:

Response:

Thank you. We made amendment.

Our inclusion criteria were as follows: (1) unilateral shoulder pain lasting more than 1 month, (2) restriction of shoulder range of motion in the coronal, sagittal, and frontal plane. and (3) age > 20 years old.

Comment 8: Materials and Methods- Subject

- "Was there a specific age group included in the criteria for participation?" (Children/elders etc.).

Response:

Thank you! We add this age criteria in the materials and methods. We recruited patients above 20 years old.

(3) age > 20 years old.

Comment 9: Materials and Methods- Subject

- Exclusion criteria were as followings:   The exclusion criteria were as follows:

Response:

Thank you. We made amendment.

The exclusion criteria were as follows: : (1) patients with bilateral frozen shoulders; (2) upper limb or shoulder fracture; and (3) documented rotator cuff tear, and (4) Individuals with shoulder surgery, infection, rheumatic disease, and malignancy.

Comment 10: Materials and Methods- Subject

Were individuals with shoulder surgery, infection, rheumatic disease, malignancy, or chronic diseases such as diabetes and thyroid disease included or excluded from the study?

Response:

Thank you!

(4) Individuals with shoulder surgery, infection, rheumatic disease, malignancy were excluded from this study.

Comment 11: Materials and Methods- Subject

This retrospective study included 362 ??? patients. (The total of the groups is 363).

Response:

This retrospective study included 362 patients. We made a typo in the 2.1 Subject, (Group 2, N=53 instead of N=54). Therefore, the total of the groups is 362

Frozen shoulder without involvement of the scalene complex and levator scapulae was denoted Group 1 (N=254), versus frozen shoulder with involvement of the scalene complex (Group 2, N=53) or levator scapulae (Group 3, N=46) and frozen shoulder with involvement of both muscles (Group 4, N=9).

Comment 12: Materials and Methods- Subject

- ..............from the patients' electronic... ; .......from the patient's electronic..... 

- ............while patient seated with.......; ........while the patient was seated.....

Response:

Thank you. We made amendment.

were obtained from the patient’s electronic medical records.

Patients’ maximum painless range of shoulder motion(including flexion, abduction, and external rotation)was  measured using a goniometer while the patient was seated with their clavicle and scapular fixed.

Comment 13: Materials and Methods- Ultrasound Examination

Could you please provide information about the number of years the doctor has been performing shoulder ultrasounds?

Response:

The ultrasound examination was done by a single doctor who has been performing shoulder ultrasounds for more than 20 years.

Comment 14: Materials and Methods- Ultrasound Examination

- ....the SS ???? tendon....  (Let's write it short using parentheses).

Response:

Thank you. We made amendment.

in order to check the supraspinatus(SS) tendon and subdeltoid subacromial bursa.

Comment 15: Materials and Methods- Ultrasound Examination

- Let's add small boxes next to Figures 1 and 2 to show the areas where we placed the probe.

Response:

Yes, we add the position of probe next to Figures 1 and 2.

Position of the probe was shown in the smaller frame on the right hand side next to Figures 1 and 2.

Comment 16: Materials and Methods- Statistical Analysis

- In some outcomes, such as age (mean ± SD) was not used? (SD; Standard deviation)

Response:

Thank you. We made amendment.

The cohort included 159 males and 203 females with an average age of 58.2±13.2 (from 25 to 99 years old).

Comment 17: Results

- Female was predominant.;  Females were predominant.

- Table 1. Patients’ characteristics, risk factor of ..., ; Patients’ characteristics, the risk factors of ...

Response:

Thank you. We made amendment.

Females were predominant within our cohort.

Comment 19: Results

- Let's write the abbreviations below the table. (DM, MACC, COPD......).     Let's write SD instead of std.

Response:

Thank you. We made amendment.

Abbreviations: DM: Diabetes Mellitus, MACC: Major Adverse Cardiovascular Events, COPD: Chronic Obstructive Pulmonary Disease

Comment 18: Results

- Table 1.    The regulation of age/years  should be considered;

Age<50,

65<Age>51,

85<Age>66,           Please make sure to arrange the table accordingly.

Age>80.

Response:

Thank you. We made amendment.

Table 1. Patients’ characteristics, risk of adhesive capsulitis, and shoulder range of motion.

Non (=254)

Levator scapulae

(=46)

Scalene complex

(=53)

Both (=9)

P-value

N

%

N

%

N

%

N

%

Sex

0.8875

FEMALE

140

55.1%

27

58.7%

30

56.6%

6

66.7%

MALE

114

44.9%

19

41.3%

23

43.4%

3

33.3%

Age_GROUP

0.7910

AGE<50

61

24.0%

13

28.3%

19

35.8%

3

33.3%

AGE 51-65

104

40.9%

18

39.1%

20

37.7%

4

44.4%

AGE 66-85

75

29.5%

13

28.3%

11

20.8%

1

11.1%

AGE>85

14

5.5%

2

4.3%

3

5.7%

1

11.1%

Smoking

24

9.4%

2

4.3%

4

7.5%

0

0.0%

0.5238

Alcohol

9

3.5%

2

4.3%

2

3.8%

0

0.0%

0.9365

Rheumatoid arthritis (RA)

4

1.6%

0

0.0%

2

3.8%

0

0.0%

0.4349

Thyroid problems

3

1.2%

3

6.5%

4

7.5%

0

0.0%

0.0227

DM

41

16.1%

1

2.2%

7

13.2%

2

22.2%

0.0784

GOUT

16

6.3%

2

4.3%

1

1.9%

1

11.1%

0.5133

Hypertension

71

28.0%

8

17.4%

11

20.8%

2

22.2%

0.3793

Hyperlipidemia

57

22.4%

7

15.2%

10

18.9%

2

22.2%

0.7059

MACC

15

5.9%

1

2.2%

4

7.5%

1

11.1%

0.5993

Spondylosis

51

20.1%

11

23.9%

13

24.5%

1

11.1%

0.7306

COPD

32

12.6%

9

19.6%

2

3.8%

1

11.1%

0.1161

MEAN

SD

MEAN

SD

MEAN

SD

MEAN

SD

Flexion

121

28.82

144

15.94

138

19.75

142

28.17

<.0001

Abduction

52

13.88

61

14.88

58

12.17

59

15.57

<.0001

External rotation

31

17.34

40

17.17

39

13.44

42

19.53

0.0003

Total ROM

203

50.02

244

36.29

236

34.43

242

53.04

<.0001

Abbreviations: DM: Diabetes Mellitus, MACC: Major Adverse Cardiovascular Events, COPD: Chronic Obstructive Pulmonary Disease

Comment 20: Results

- 144±15.84, let's correct it. (144±15.94)

Response:

Thank you. We made amendment.

in Group 1; 144±15.94, 61±14.88, 40±17.17 and 244±36.29 degrees, respectively,

Comment 21: Results

- Table 2. Fix age.

Response:

Thank you. We made amendment.

Comment 22: Results

- Table 2. What does Amoking mean?

Response:

Thank you. We made amendment.

Table 2. Relationship between total shoulder ROM and other clinical data.

Total_ROM

R2

19.3%

REF

Estimated value

95%CI

P-value

Group 2: Scalene complex

Group 1

32.3906

18.7889

45.9923

<.0001

Group 3: Levator scapulae

38.2534

23.876

52.6307

<.0001

Group 4: Both

38.9318

8.9646

68.8989

0.0109

Male

Female

-4.3558

-14.5771

5.8655

0.4036

AGE>85

AGE<50

-35.8777

-59.8045

-11.951

0.0033

AGE 65-85

-17.5805

-31.2241

-3.9368

0.0116

AGE 50-64

-8.8953

-20.9484

3.1577

0.148

Smoking

NO

-7.042

-26.4535

12.3695

0.4771

Alcohol

NO

-9.6668

-36.7555

17.422

0.4843

Thyroid problems

NO

4.4709

-24.7098

33.6515

0.764

DM

NO

-9.493

-26.0829

7.0969

0.2621

GOUT

NO

0.9952

-21.2726

23.263

0.9302

Hypertension

NO

0.946

-12.0346

13.9266

0.8864

Hyperlipidemia

NO

14.9299

-0.4874

30.3473

0.0577

Rheumatoid arthritis

NO

-13.2946

-43.764

17.1748

0.3925

MACC

NO

-24.9725

-46.5906

-3.3544

0.0236

Spondylosis

NO

0.8853

-11.436

13.2066

0.888

COPD

NO

6.5472

-8.1451

21.2395

0.3824

Abbreviations: DM: Diabetes Mellitus, MACC: Major Adverse Cardiovascular Events, COPD: Chronic Obstructive Pulmonary Disease

Comment 23: Results

- Table 2. Let's write the abbreviations below the table. (DM, MACC, COPD......)

Response:

Thank you. We made amendment.

Abbreviations: DM: Diabetes Mellitus, MACC: Major Adverse Cardiovascular Events, COPD: Chronic Obstructive Pulmonary Disease

Comment 24: Discussion

- The findings from Rundquist P. J. group [10].....; The findings from Rundquist et al. [10]

- Vermeulen [11] et al; Vermeulen et al. [11]

Response:

Thank you. We made amendment.

The findings from Rundquist PJ et al. [28] support the theory of scapular compensation after loss of glenohumeral ROM for the purpose of achieving greater humerus-to-trunk scapular plane elevation. Vermeulen et al. [29] also demonstrated the increased contribution of the scapular muscles to scapulohumeral rhythm in subjects with frozen shoulder.

Comment 25: Discussion

- Previous studies have indicated that levator scapulae is activated upon shoulder elevation.  (Let's add a references).

Response:

We added the reference.

Poland J,Hobart DJ,Payton. The Musculosketal System Second Edition. Medical Examination Publishing Co., Inc., New York.Page 127.

Comment 26: Discussion

- The discussion focuses more on pathophysiology than on the US. Let's reorganize it by emphasizing more emphasis on US studies.

Response:

We added some more description on US studies

Recently, with the advancement of interventional musculoskeletal ultrasound, those lesions in frozen shoulder can be detected by ultrasound. Therefore, echo-guided injection to those lesions became possible with steroid or other agents such as platelet rich plasm. Exercise program can be started after inflammation have been well controlled.  

Comment 27: Conclusion

- Are your clinic's scalene muscles and levator scapulae examined during routine shoulder-US? (Because the study is retrospective).

Response:

After we found the importance of these two muscles about 8 years ago. Scalene muscles and levator scapulae were examined during routine shoulder-US.

After we found the importance of these two muscles about 8 years ago. Scalene muscles and levator scapulae were examined during routine shoulder-US.

Comment 28: Conclusion

- The similarity rate is 37%.  It should be reduced.

Response:

Thank you. We reduced the similarity in the revised manuscript.

Comment 29: Reference

References must be up to date.

References need to be rearranged.

Response:

Thank you. We made amendment.

We have rearranged all the references.

References

  1. Hannafin, J.A.; Chiaia, T.A. Adhesive capsulitis. A treatment approach. Clin Orthop Relat Res 2000, 95-109.
  2. Xiao, R.C.; DeAngelis, J.P.; Smith, C.C.; Ramappa, A.J. Evaluating Nonoperative Treatments for Adhesive Capsulitis. J Surg Orthop Adv 2017, 26, 193-199.
  3. Mengiardi, B.; Pfirrmann, C.W.; Gerber, C.; Hodler, J.; Zanetti, M. Frozen shoulder: MR arthrographic findings. Radiology 2004, 233, 486-492, doi:10.1148/radiol.2332031219.
  4. Li, J.Q.; Tang, K.L.; Wang, J.; Li, Q.Y.; Xu, H.T.; Yang, H.F.; Tan, L.W.; Liu, K.J.; Zhang, S.X. MRI findings for frozen shoulder evaluation: is the thickness of the coracohumeral ligament a valuable diagnostic tool? PLoS One 2011, 6, e28704, doi:10.1371/journal.pone.0028704.
  5. Zhao, W.; Zheng, X.; Liu, Y.; Yang, W.; Amirbekian, V.; Diaz, L.E.; Huang, X. An MRI study of symptomatic adhesive capsulitis. PLoS One 2012, 7, e47277, doi:10.1371/journal.pone.0047277.
  6. Carrillon, Y.; Noel, E.; Fantino, O.; Perrin-Fayolle, O.; Tran-Minh, V.A. Magnetic resonance imaging findings in idiopathic adhesive capsulitis of the shoulder. Rev Rhum Engl Ed 1999, 66, 201-206.
  7. Sofka, C.M.; Ciavarra, G.A.; Hannafin, J.A.; Cordasco, F.A.; Potter, H.G. Magnetic resonance imaging of adhesive capsulitis: correlation with clinical staging. Hss j 2008, 4, 164-169, doi:10.1007/s11420-008-9088-1.
  8. Fields, B.K.K.; Skalski, M.R.; Patel, D.B.; White, E.A.; Tomasian, A.; Gross, J.S.; Matcuk, G.R., Jr. Adhesive capsulitis: review of imaging findings, pathophysiology, clinical presentation, and treatment options. Skeletal Radiol 2019, 48, 1171-1184, doi:10.1007/s00256-018-3139-6.
  9. Homsi, C.; Bordalo-Rodrigues, M.; da Silva, J.J.; Stump, X.M. Ultrasound in adhesive capsulitis of the shoulder: is assessment of the coracohumeral ligament a valuable diagnostic tool? Skeletal Radiol 2006, 35, 673-678, doi:10.1007/s00256-006-0136-y.
  10. Lho, Y.M.; Ha, E.; Cho, C.H.; Song, K.S.; Min, B.W.; Bae, K.C.; Lee, K.J.; Hwang, I.; Park, H.B. Inflammatory cytokines are overexpressed in the subacromial bursa of frozen shoulder. J Shoulder Elbow Surg 2013, 22, 666-672, doi:10.1016/j.jse.2012.06.014.
  11. Bunker, T.D.; Reilly, J.; Baird, K.S.; Hamblen, D.L. Expression of growth factors, cytokines and matrix metalloproteinases in frozen shoulder. J Bone Joint Surg Br 2000, 82, 768-773, doi:10.1302/0301-620x.82b5.9888.
  12. Kabbabe, B.; Ramkumar, S.; Richardson, M. Cytogenetic analysis of the pathology of frozen shoulder. Int J Shoulder Surg 2010, 4, 75-78, doi:10.4103/0973-6042.76966.
  13. Marchese, C.; Cristalli, G.; Pichi, B.; Manciocco, V.; Mercante, G.; Pellini, R.; Marchesi, P.; Sperduti, I.; Ruscito, P.; Spriano, G. Italian cross-cultural adaptation and validation of three different scales for the evaluation of shoulder pain and dysfunction after neck dissection: University of California - Los Angeles (UCLA) Shoulder Scale, Shoulder Pain and Disability Index (SPADI) and Simple Shoulder Test (SST). Acta Otorhinolaryngol Ital 2012, 32, 12-17.
  14. Fayad, F.; Roby-Brami, A.; Yazbeck, C.; Hanneton, S.; Lefevre-Colau, M.M.; Gautheron, V.; Poiraudeau, S.; Revel, M. Three-dimensional scapular kinematics and scapulohumeral rhythm in patients with glenohumeral osteoarthritis or frozen shoulder. J Biomech 2008, 41, 326-332, doi:10.1016/j.jbiomech.2007.09.004.
  15. Ludewig, P.M.; Cook, T.M. Alterations in shoulder kinematics and associated muscle activity in people with symptoms of shoulder impingement. Phys Ther 2000, 80, 276-291.
  16. Lin, J.J.; Hanten, W.P.; Olson, S.L.; Roddey, T.S.; Soto-quijano, D.A.; Lim, H.K.; Sherwood, A.M. Functional activity characteristics of individuals with shoulder dysfunctions. J Electromyogr Kinesiol 2005, 15, 576-586, doi:10.1016/j.jelekin.2005.01.006.
  17. Hsu, Y.H.; Chen, W.Y.; Lin, H.C.; Wang, W.T.; Shih, Y.F. The effects of taping on scapular kinematics and muscle performance in baseball players with shoulder impingement syndrome. J Electromyogr Kinesiol 2009, 19, 1092-1099, doi:10.1016/j.jelekin.2008.11.003.
  18. Shih YF, Liao PW, Lee CS. The immediate effect of muscle release intervention on muscle activity and shoulder kinematics in patients with frozen shoulder: a cross-sectional, exploratory study. BMC Musculoskelet Disord. 2017 Nov 28;18(1):499. doi: 10.1186/s12891-017-1867-8. PMID: 29183307; PMCID: PMC5706296.
  19. Shih, Y.F.; Liao, P.W.; Lee, C.S. The immediate effect of muscle release intervention on muscle activity and shoulder kinematics in patients with frozen shoulder: a cross-sectional, exploratory study. BMC Musculoskelet Disord 2017, 18, 499, doi:10.1186/s12891-017-1867-8.
  20. Zhao W, Zheng X, Liu Y, Yang W, Amirbekian V, Diaz LE, et al. An MRI study of symptomatic. adhesive capsulitis. PLoS One. 2012;7:e47277. doi: 10.1371/journal.pone.0047277.
  21. Mao CY, Jaw WC, Cheng H. Frozen shoulder: correlation between the response to physical therapy and follow-up shoulder arthrography. Arch Phys Med Rehabil. 1997;78:857–859. doi: 10.1016/S0003-9993(97)90200-8.
  22. Hung CJ, Hsieh CL, Yang PL, Lin JJ. Relationships between posterior shoulder muscle stiffness and rotation in patients with stiff shoulder. J Rehabil Med. 2010 Mar;42(3):216-20. doi: 10.2340/16501977-0504. PMID: 20411215.
  23. Falla D. and Neuromuscular control of the cervical spine in neck pain disorders. In:  Graven-Nielsen T, , Arendt-Nielsen L, and Mense S, eds.  Fundamentals of Musculoskeletal Pain. Seattle, WA: IASP Press;  2008: 417– 430. ;
  24. Falla D, , Bilenkij G, , Jull G. and Patients with chronic neck pain erns of muscle activation during performance of a functional upper limb task.  Spine (Phila Pa 1976).  2004; 29: 1436– 1440.
  25. Falla DL, , Jull GA, , Hodges PW. and Patients with neck pain demonstrate reduced electromyographic activity of the deep cervical flexor muscles during performance of the craniocervical flexion test.  Spine (Phila Pa 1976).  2004; 29: 2108– 2114.
  26. Madeleine P, , Lundager B, , Voigt M, , Arendt-Nielsen L. and Shoulder muscle co-ordination during chronic and acute experimental neck-shoulder pain. An occupational pain study.  Eur J Appl Physiol Occup Physiol.  1999; 79: 127– 140.
  27. Hidalgo-Lozano, A.; Calderón-Soto, C.; Domingo-Camara, A.; Fernández-de-Las-Peñas, C.; Madeleine, P.; Arroyo-Morales, M. Elite swimmers with unilateral shoulder pain demonstrate altered pattern of cervical muscle activation during a functional upper-limb task. J Orthop Sports Phys Ther 2012, 42, 552-558, doi:10.2519/jospt.2012.3875.
  28. Rundquist, P.J. Alterations in scapular kinematics in subjects with idiopathic loss of shoulder range of motion. J Orthop Sports Phys Ther 2007, 37, 19-25, doi:10.2519/jospt.2007.2121.
  29. Vermeulen, H.M.; Stokdijk, M.; Eilers, P.H.; Meskers, C.G.; Rozing, P.M.; Vliet Vlieland, T.P. Measurement of three dimensional shoulder movement patterns with an electromagnetic tracking device in patients with a frozen shoulder. Ann Rheum Dis 2002, 61, 115-120, doi:10.1136/ard.61.2.115.
  30. Poland J,Hobart DJ,Payton. The Musculosketal System Second Edition. Medical Examination Publishing Co., Inc., New York.Page 127.
  31. De Freitas, V.; Vitti, M.; Furlani, J. Electromyographic study of levator scapulae and rhomboideus major muscles in movements of the shoulder and arm. Electromyogr Clin Neurophysiol 1980, 20, 205-216.
  32. Castelein, B.; Cagnie, B.; Parlevliet, T.; Cools, A. Superficial and Deep Scapulothoracic Muscle Electromyographic Activity During Elevation Exercises in the Scapular Plane. J Orthop Sports Phys Ther 2016, 46, 184-193, doi:10.2519/jospt.2016.5927.
  33. Castelein, B.; Cagnie, B.; Parlevliet, T.; Cools, A. Scapulothoracic muscle activity during elevation exercises measured with surface and fine wire EMG: A comparative study between patients with subacromial impingement syndrome and healthy controls. Man Ther 2016, 23, 33-39, doi:10.1016/j.math.2016.03.007.

  1. Ferlito R, Testa G, McCracken KL, et al. Effectiveness of Therapeutical Interventions on the Scapulothoracic Complex in the Management of Patients with Subacromial Impingement and Frozen Shoulder: A Systematic Review. J Funct Morphol Kinesiol. 2023, 8(2):38. doi:10.3390/jfmk8020038
  2. Hidalgo-Lozano, A.; Calderón-Soto, C.; Domingo-Camara, A.; Fernández-de-Las-Peñas, C.; Madeleine, P.; Arroyo-Morales, M. Elite swimmers with unilateral shoulder pain demonstrate altered pattern of cervical muscle activation during a functional upper-limb task. J Orthop Sports Phys Ther 2012, 42, 552-558, doi:10.2519/jospt.2012.3875.
  3. Madeleine, P.; Lundager, B.; Voigt, M.; Arendt-Nielsen, L. Shoulder muscle co-ordination during chronic and acute experimental neck-shoulder pain. An occupational pain study. Eur J 24.Behrsin, J.F.; Maguire, K. Levator Scapulae Action during Shoulder Movement: A Possible Mechanism for Shoulder Pain of Cervical Origin. Aust J Physiother 1986, 32, 101-106, doi:10.1016/s0004-9514(14)60646-2.
  4. Akbar M, McLean M, Garcia-Melchor E, Crowe LA, McMillan P, Fazzi UG, et al. Fibroblast activation and inflammation in frozen shoulder. PLoS ONE. (2019) 14:e0215301. doi: 10.1371/journal.pone.0215301
  5. Simmonds RE, Foxwell BM. Signalling, inflammation and arthritis: NF-κ B and its relevance to arthritis and inflammation. Rheumatology. (2008) 47:584–90. doi: 10.1093/rheumatology/kem298.
  6. de la Serna D, Navarro-Ledesma S, Alayón F, López E, Pruimboom L. A Comprehensive View of Frozen Shoulder: A Mystery Syndrome. Front Med (Lausanne). 2021 May 11;8:663703. doi: 10.3389/fmed.2021.663703. PMID: 34046418; PMCID: PMC8144309. ) Apart from famous inflammatory cascade mediated by the transcription factor, nuclear factor Kappa B (NF- kB)
  7. Quiñonez-Flores CM, González-Chávez SA, Pacheco-Tena C. Hypoxia and its implications in rheumatoid arthritis. J Biomed Sci. (2016) 23:62. doi: 10.1186/s12929-016-0281-0.
  8. Wang N, Liang H, Zen K . Molecular mechanisms that influence the macrophage m1-m2 polarization balance. Front Immunol 2014; 5: 614.
  9. Sica A, Mantovani A . Macrophage plasticity and polarization: in vivo veritas. J Clin Invest 2012; 122: 787–795.
  10. Liu, T., Zhang, L., Joo, D. et al. NF-κB signaling in inflammation. Sig Transduct Target Ther 2, 17023 (2017). https://doi.org/10.1038/sigtrans.2017.23.
  11. Cher, J.Z.B.; Akbar, M.; Kitson, S.; Crowe, L.A.N.; Garcia-Melchor, E.; Hannah, S.C.; McLean, M.; Fazzi, U.G.; Kerr, S.C.; Murrell, G.A.C.; et al. Alarmins in Frozen Shoulder: A Molecular Association Between Inflammation and Pain. Am J Sports Med 2018, 46, 671-678, doi:10.1177/0363546517741127.
  12. Chen, R.; Kang, R.; Tang, D. The mechanism of HMGB1 secretion and release. Exp Mol Med 2022, 54, 91-102, doi:10.1038/s12276-022-00736-w.
  13. de la Serna, D.; Navarro-Ledesma, S.; Alayón, F.; López, E.; Pruimboom, L. A Comprehensive View of Frozen Shoulder: A Mystery Syndrome. Front Med (Lausanne) 2021, 8, 663703, doi:10.3389/fmed.2021.663703.
  14. Hand, G.C.; Athanasou, N.A.; Matthews, T.; Carr, A.J. The pathology of frozen shoulder. J Bone Joint Surg Br 2007, 89, 928-932, doi:10.1302/0301-620x.89b7.19097.

Comment 30:

Comments on the Quality of English Language

English editing is needed.

Response:

Thank you. We made amendment.

English editing had completed.
